# On Training-Conditional Conformal Prediction and Binomial Proportion Confidence Intervals

**Rudi Coppola**                                                        *r.coppola@tudelft.nl*
*Department of Mechanical Engineering*
*Delft University of Technology*

**Manuel Mazo Jr.**                                                     *m.mazo@tudelft.nl*
*Department of Mechanical Engineering*
*Delft University of Technology*

**Reviewed on OpenReview:** *https://openreview.net/forum?id=pSk5qyt1ob*

## Abstract

Estimating the expectation of a Bernoulli random variable based on $N$ independent trials is a classical problem in statistics, typically addressed using Binomial Proportion Confidence Intervals (BPCI). In the control systems community, many critical tasks—such as certifying the statistical safety of dynamical systems—can be formulated as BPCI problems.

Conformal Prediction (CP), a distribution-free technique for uncertainty quantification, has gained significant attention in recent years and has been applied to various control systems problems, particularly to address uncertainties in learned dynamics or controllers. A variant known as *training-conditional CP* was recently employed to tackle the problem of safety certification.

In this note, we highlight that the use of training-conditional CP in this context does not provide valid safety guarantees. We demonstrate why CP is unsuitable for BPCI problems and argue that traditional BPCI methods are better suited for statistical safety certification.

## 1 Introduction

Uncertainty quantification is a critical aspect in fields where predictions influence safety and performance guarantees, such as in control systems. Probabilistic guarantees, including those derived from the theory of Probably Approximately Correct (PAC) learning, play an important role in providing bounds on the accuracy of predictions under limited training data.

Conformal Prediction (CP) is one of the approaches that has gained visibility due to its ability to provide valid prediction sets without requiring strong distributional assumptions. A distinctive characteristic of CP is that, rather than providing a point prediction of the variable of interest, it provides set predictions with a valid bound on the probability that the predicted set contains the true variable (Vovk et al., 2005). This technical note focuses on a specific formulation of CP known as training-conditional CP (Vovk, 2012). However, existing applications in areas such as safety verification for dynamical systems have shown limitations in the interpretation of these guarantees. In particular, recent works have applied training-conditional CP to safety verification problems in control systems (Chilakamarri et al., 2024; Lin & Bansal, 2024; Vincent et al., 2024). While promising, these applications have misinterpreted the implications of CP's set prediction framework, especially in cases where the underlying data can be modeled as Bernoulli random variables. This paper aims to rigorously analyze these limitations and provide an alternative framework for interpreting PAC-based guarantees in such contexts. In section 2 we recall existing methods for estimating the expectation of a Bernoulli random variable. In section 3 we introduce the formalism of training-conditional CP, followed by a detailed analysis of its PAC guarantees. In section 4 we present a special case of interest, where the nonconformity measure corresponds to an indicator function, leading to Bernoulli-distributed conformity

scores. We demonstrate that the PAC guarantees derived from this setting are unsuitable for estimating the expectation of a Bernoulli random variable.

## 2 Binomial Proportion Confidence Intervals

Consider a setting where there are $N + 1$ independent and identically distributed (i.i.d.) Bernoulli random variables (r.v.) $R_1, R_2, ..., R_N, R_{N+1}$ with parameter $b$, i.e. $R_i \sim \text{Bern}_b$ and $\text{Pr}_{\text{Bern}_b}(R_i = 1) \doteq b$. Given a realization of the first $N$ r.v., the problem is to estimate an interval of values for the probability that the $N + 1$-th variable will be equal to 1, or in other words we want to estimate the parameter $b$. This is a very well studied problem and it is known in the literature under the name of Binomial Proportion Confidence Intervals (BPCI), see Dean & Pagano (2015) for a survey. We give below a quick overview of the setting.

Define the new r.v. $Y \doteq \sum_{i=1}^{N} R_i$. It is well known that $Y$ has a binomial distribution $Y \sim \text{Bin}_{N,b}$ with $N$ trials and probability of success $b$, defined by $\text{Pr}_{\text{Bin}_{N,b}}(Y = y) \doteq \binom{N}{y} b^y (1-b)^{N-y}$ for $y \in \mathbb{Z}_{[0,N]}$, where $\mathbb{Z}_{[0,N]}$ denotes the integers $0, 1, ..., N$. Let $\check{b} : \mathbb{Z}_{[0,N]} \to [0,1]$ and $\hat{b} : \mathbb{Z}_{[0,N]} \to [0,1]$ be two random variables serving as interval estimators. The coverage probability of the interval estimator $[\check{b}, \hat{b}]$ for $Y \sim \text{Bin}_{N,b}$ is defined as

$$\rho(b, \check{b}, \hat{b}) \doteq \text{Pr}_{\text{Bin}_{N,b}}(\check{b}(Y) \le b \le \hat{b}(Y)). \tag{1}$$

In the expression above $b$ is fixed and it's the true parameter of the binomial distribution describing $Y$. Note that $\check{b}$ and $\hat{b}$ are a transformation of the same random variable $Y$. This expression can also be rewritten equivalently as

$$\rho(b, \check{b}, \hat{b}) = \sum_{y \in I} \text{Pr}_{\text{Bin}_{N,b}}(Y = y),$$

where $I \doteq \{y \in \mathbb{Z}_{[0,N]} : \check{b}(y) \le b \le \hat{b}(y)\}$. For $\alpha \in (0,1)$ an interval estimator $[\check{b}, \hat{b}]$ is a *conservatively valid* (sometimes also called 'exact' or 'secure') $1 - \alpha$ confidence interval if the coverage probability $\rho(b, \check{b}, \hat{b})$ is greater or equal to $1 - \alpha$ for all the values of $b$. An example of a conservatively valid interval estimator is given by the Clopper-Pearson method (Clopper & Pearson, 1934), see also Dean & Pagano (2015) for more estimators.

Before concluding this section, we rewrite equation 1 in an equivalent form that is more commonly found in the literature on Probably Approximately Correct (PAC) bounds. First, note that $\text{Pr}_{\text{Bern}_b}(R_{N+1} = 1) = b$. Second, since $Y$ is a r.v. obtained as a transformation of the i.i.d. Bernoulli random variables $R_1, \ldots, R_N$, the probability of any event $M \subseteq \mathbb{Z}_{[0,N]}$, $\text{Pr}_{\text{Bin}_{N,b}}(Y \in M)$ can be equivalently described by $\text{Pr}_{\text{Bern}_b}^N(\{(r_1, ..., r_N) : \sum_{i \le N} r_i \in M\})$, where $\text{Pr}_{\text{Bern}_b}^N$ is the product probability measure induced by the $N$ i.i.d. Bernoulli random variables. Hence, we rewrite the definition of a conservatively valid $1 - \alpha$ confidence interval by revisiting equation 1:

$$\text{Pr}_{\text{Bin}_{N,b}}(\check{b}(Y) \le b \le \hat{b}(Y)) = \text{Pr}_{\text{Bern}_b}^N \left( \check{b}\left(\sum_{i \le N} R_i\right) \le \text{Pr}_{\text{Bern}_b}(R_{N+1} = 1) \le \hat{b}\left(\sum_{i \le N} R_i\right) \right) \ge 1 - \alpha, \tag{2}$$

for all $b \in [0,1]$. We will use this form of the coverage probability to draw a comparison with the guarantees given by training-conditional CP.

## 3 Training-conditional Conformal Prediction

Conformal Prediction is a statistical tool that uses the available data sampled from identically and independently from an underlying distribution to output predictions for which an error probability can be computed. The original formulation of CP can be informally explained as follows. Suppose that we want to solve a classification problem and we have method that given a feature $x$ outputs a label $\hat{y}$. Given a desired error probability $\epsilon$, conformal prediction uses the available data to generate a *set of labels*, typically containing $\hat{y}$, containing the true label $y$ corresponding to the feature $x$ with a probability not smaller than $1 - \epsilon$ (Shafer & Vovk, 2008). It is a method capable of augmenting a (usually unreliable) point prediction to a set prediction

with probabilistic guarantees of correctness, i.e. it construct a *set predictor*. The original formulation of CP has been successfully applied to both classification and regression problems, see Angelopoulos et al. (2023); Fontana et al. (2023) for a recent survey.

In this section we introduce instead the basic concepts of *training-conditional* CP (Vovk, 2012). Training-conditional CP is a variant of the original formulation of CP. While the quality of the guarantees differs from the original, the core idea remains the same, that is, constructing set predictions with some form of guarantees: training-conditional CP produces PAC-style guarantees. In the following, we give a self-contained overview of the theoretical details of training-conditional CP.

Let $(\mathbf{Z}, \mathcal{F}, \mathbb{P})$ be a probability space where $\mathbf{Z}$, $\mathcal{F}$ and $\mathbb{P}$ denote a sample set, a $\sigma$-algebra, and a probability measure respectively, and consider $L + 1$ i.i.d. random variables (r.v.) $Z'_1, ..., Z'_M, Z_1, ..., Z_N$ and $Z_{N+1}$ with $L = N + M$. Let $Z'_i$ for $i = 1, ..., M$ be the *training set* and $Z_i$ for $i = 1, ..., N$ be the *calibration set*. Note that $Z_{N+1}$ is not part of either set. We use the lower case of a r.v. to denote a realization[1]. An *Inductive Nonconformity M-measure* (INM) is a measurable function $A : \mathbf{Z}^M \times \mathbf{Z} \to \mathbb{R}$. While no additional requirements are needed for $A$, intuitively an effective INM will assign a high real number to any element in $\mathbf{Z}$ that does not conform to a training set (in $\mathbf{Z}^M$). An *Inductive Nonconformal Predictor* (INP) is a set predictor defined as

$$\Gamma^\epsilon(z_1, ..., z_N, z'_1, ..., z'_M) \doteq \{z \in \mathbf{Z} \; : \; p^z > \epsilon\}, \tag{3}$$

where $\epsilon \in [0, 1]$ is the *significance level*, the $p$-values are defined as

$$p^z \doteq \frac{|\{i : R_i \geq R^z\}| + 1}{N + 1}, \tag{4}$$

and

$$R_i \doteq A((z'_1, ..., z'_M), z_i) \text{ for } i = 1, ...N, \qquad R^z \doteq A((z'_1, ..., z'_M), z), \tag{5}$$

are the *nonconformity scores*.

In the following, when it is clear from the context we omit the arguments of the INP and write $\Gamma^\epsilon$ instead of $\Gamma^\epsilon(z_1, ..., z_N, z'_1, ..., z'_M)$. Intuitively, $z$ belongs to the INP $\Gamma^\epsilon$ if there are strictly more than $\lfloor \epsilon(N+1) - 1 \rfloor$ elements $R_i$ in the calibration set with a higher (worse) or equal nonconformity score than $R^z$. It is easy to see that $\epsilon' < \epsilon''$ implies that $\Gamma^{\epsilon''} \subseteq \Gamma^{\epsilon'}$. The INP is the set predictor mentioned in the discussion at the beginning of this section: similarly to the original formulation of CP, given some prediction method depending on the training set, the INP uses the available calibration set to produce a set prediction guaranteed to contain the correct prediction. The elements included in the set prediction are all the $z \in \mathbf{Z}$ that conform well enough with the calibration set, according to the chosen INM. The following theorem specifies the PAC-style guarantees for training-conditional CP.

**Theorem 1 (Vovk (2012))** *Choose $\epsilon, E \in [0, 1]$[2], fix the training set $Z'_1 = z'_1, ..., Z'_M = z'_M$, let $N$ be the size of the calibration set, and consider the event*

$$S_E \doteq \{(z_1, ..., z_N) \in \mathbf{Z}^N : \mathbb{P}(Z_{N+1} \in \Gamma^\epsilon(z_1, ..., z_N, z'_1, ..., z'_M)) \geq 1 - E\} \tag{6}$$

*in the $\sigma$-algebra $\mathcal{F}^N$ of the product probability space $(\mathbf{Z}^N, \mathcal{F}^N, \mathbb{P}^N)$, where $\Gamma^\epsilon$ is defined according to equations 3-5. It holds that*

$$\mathbb{P}^N(S_E) \geq 1 - \delta, \tag{7}$$

*where $\delta \doteq Bin_{N,E}(J) = \sum_{j=0}^{J} \binom{N}{j} E^j (1-E)^{N-j}$ is the cumulative binomial distribution with $N$ trials and probability of success $E$, with $J \doteq \lfloor \epsilon(N+1) - 1 \rfloor$.*

---

[1]Our considerations hold also for the case where $\mathbf{Z} = \mathbf{X} \times \mathbf{Y}$ where $\mathbf{X}$ and $\mathbf{Y}$ represent a measurable feature space and label space respectively and each $z \in \mathbf{Z}$ may be written as $z = (x, y)$ where $x \in \mathbf{X}$ is some feature and $y \in \mathbf{Y}$ a label. For clarity we omit the exact structure of $\mathbf{Z}$.

[2]A brief note on the notation. In the original formulation of CP $\epsilon$ has a double role: it is the significance level (appearing as the index to the INP $\Gamma^\epsilon$) and it describes the coverage probability as $1 - \epsilon$, see Shafer & Vovk (2008) for details. In the training-conditional formulation the latter role is covered by $E$, that is $1 - E$ is the coverage probability and $\epsilon$ remains the significance level, see Vovk (2012).

The quantities $1 - \delta$ and $1 - E$ are sometimes referred to as the *confidence* and *coverage probability* (which is not the coverage probability mentioned in section 2). Theorem 1 is to be understood in the following way. Given two values $\epsilon$ and $E$, for the given training set, the event $S_E$ is the subset of $\mathbf{Z}^N$ containing all the tuples $(z_1, ..., z_N)$ such that the INP $\Gamma^\epsilon$ contains a realization of $Z_{N+1}$ with probability at least $1 - E$, or, in other words, $\Gamma^\epsilon$ returns a subset of $\mathbf{Z}$ of measure at least $1 - E$. By *equation* 7 the measure of this set of tuples $S_E$ is at least $1 - \delta$, where $\delta$ depends on $\epsilon$, $\beta$ and $N$. This form of guarantees where a double layer of nested probabilities is present is called Probably Approximately Correct (PAC). Moreover, this is a distribution-free result, that is, it holds for every $\mathbb{P}$ as long as the samples used to construct the INP are i.i.d. and $\mathbb{P}$-distributed. In particular, in this work we focus on Bernoulli-distributed r.v.'s, and Theorem 1 holds for any value of the parameter $b$ of a Bernoulli distribution. Observe that the confidence $1 - \delta$ and the quantity $1 - \alpha$ mentioned in section 2 play a similar role in that they described the outmost layer of probability, compare for instance equations equation 7 and equation 2. Finally, we note that $\epsilon$ and $E$ are chosen *a priori*; in other words, they cannot be defined as random variables depending on a realization of the calibration set, as is erroneously done in Lin & Bansal (2024).

## 4 A Special Case of Interest

In this section we draw a parallel between the BPCI and training-conditional CP and show the fundamental difference between the two approaches.

Let the INM be an indicator function for the set $Q \subset \mathbf{Z}$, that is

$$A((z_1', ..., z_M'), z) \doteq \begin{cases} 1 \text{ if } z \in Q, \\ 0 \text{ if } z \in \overline{Q}. \end{cases} \tag{8}$$

Typically $Q$ depends on $z_1', ..., z_M'$. For example, in binary classification problems, the training set may be used to train a parameterized function that assigns one of two labels to all $z \in Q$, as in Support Vector Machines. However, since Theorem 1 assumes a given training set, we omit this dependency here. A point $z$ with a high nonconformity score is interpreted as poorly conforming to the training set. For this reason, in section 4.1 the set $Q$ will represent the unsafe region of a dynamical system.

Given a fixed training set, the nonconformity scores of the calibration set follow an i.i.d. Bernoulli distribution with parameter $b$, i.e. $R_i \sim \text{Bern}_b$, where $b \doteq \mathbb{P}(Q)$. Using a BPCI method it is directly possible to derive a conservatively valid confidence interval for the parameter $b$ describing the probability of drawing a sample in $Q$, as shown in section 2. Can a training-conditional CP approach also provide a conservatively valid confidence interval for $b$ based on the calibration set? The answer is no. We illustrate this with an example.

**Example 1** - Part 1.

*Suppose that the calibration set has size 2, i.e. $N = 2$. Up to reindexing, there are three distinct outcomes. Case 1: With probability $(1 - b)^2$ we have $z_1, z_2 \notin Q$, resulting in nonconformity scores $R_1 = R_2 = 0$. We construct the prediction set $\Gamma^\epsilon$ following its definition equation 3.*

- *For all $z \in Q$, we have that $R^z = 1$, meaning $z$ has the highest (worst) nonconformity score. Since $|\{i \leq 2 : R_i \geq R^z\}| = 0$ the corresponding p-value is $p^z = \frac{1}{3}$.*

- *For all $z \in \overline{Q}$ we have that $R^z = 0$ resulting in and $p^z = 1$.*

*The inclusion of $z$ in the predicted set $\Gamma^\epsilon$ depends on the significance level $\epsilon$.*

- *If $\epsilon \in [\frac{1}{3}, 1)$ then any $z \in Q$ is excluded from $\Gamma^\epsilon$ since $p^z = \frac{1}{3} \leq \epsilon$, while all $z \in \overline{Q}$ are included since $p^z = 1 > \epsilon$. Thus, $\Gamma^\epsilon = \overline{Q}$.*

- *If $\epsilon \in [0, \frac{1}{3})$ then any $z \in Q \cup \overline{Q} = \mathbf{Z}$ has a sufficiently high p-value, meaning $\Gamma^\epsilon = \mathbf{Z}$.*

Case 2: *With probability $2b(1 - b)$ we have $(z_1 \in Q \wedge z_2 \in \overline{Q})$ or $(z_2 \in Q \wedge z_1 \in \overline{Q})$ hence $R_1 \cup R_2 = \{0, 1\}$.*

- *If $R^z = 1$ then $p^z = \frac{2}{3}$.*

- *If $R^z = 0$ then $p^z = 1$.*

*Thus:*

- *If $\epsilon \in [0, \frac{2}{3})$ then $\Gamma^\epsilon = \mathbf{Z}$.*

- *If $\epsilon \in [\frac{2}{3}, 1)$ then $\Gamma^\epsilon = \overline{Q}$.*

Case 3: *With probability $b^2$ we have $z_1, z_2 \in Q$ and $R_1 = R_2 = 1$.*

- *If $R^z = 1$ then $p^z = 1$.*

- *If $R^z = 0$ then $p^z = 1$ as well.*

*Then for any significance level $\epsilon \in [0, 1)$ it holds $\Gamma^\epsilon = \mathbf{Z}$.*

In summary, for any fixed $\epsilon$ the INP is fully determined by the calibration set through equations 3-5; as a result, $\Gamma^\epsilon$ can be thought equivalently as a discrete random variable with support $Q$, $\overline{Q}$ and $\mathbf{Z}$, see Figure 1.

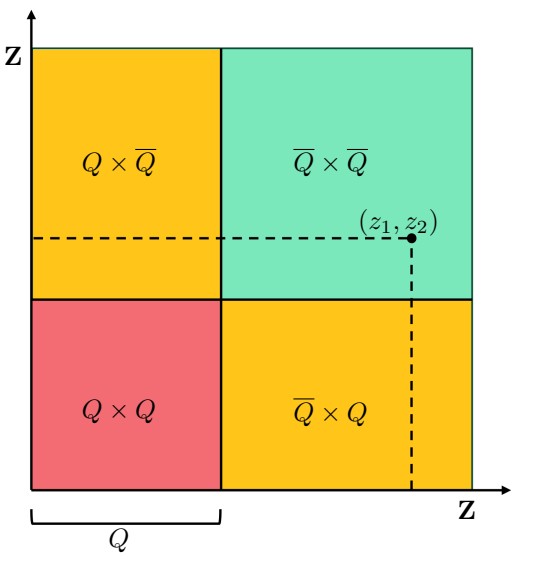 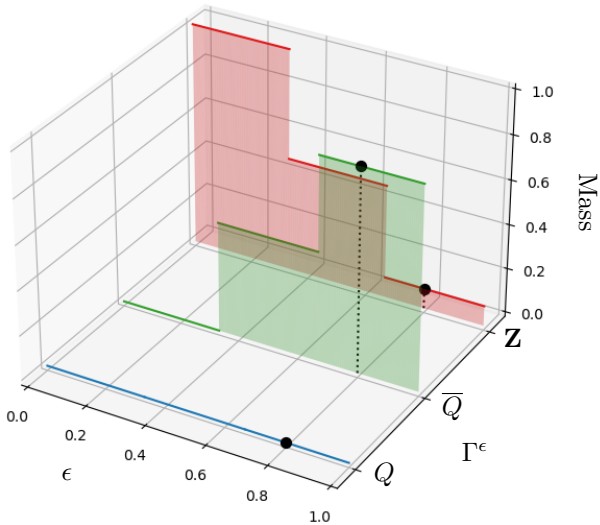

Figure 1: On the left, a representation of the product space $\mathbf{Z}^2 = \mathbf{Z} \times \mathbf{Z}$, partitioned accordingly to the sets $Q$ and $\overline{Q}$, and a hypothetical calibration set $(z_1, z_2)$ as in Case 1. On the right, a summary of Case 1, 2 and 3. On the $x$-,$y$-,$z$-axes are represented the values of $\epsilon$, the prediction (or support) of the INP, and the probability mass function respectively. For any given $\epsilon$, the INP $\Gamma^\epsilon$ can be viewed as a discrete random variable with support $Q$, $\overline{Q}$ and $\mathbf{Z}$. In the figure, for $\epsilon = 0.8$ and $b = 0.3$, the INP predicts $Q$ with probability 0, $\mathbf{Z}$ with probability $b^2$, and $\overline{Q}$ with probability $1 - b^2$.

**Example 1** - Part 2.

*Now, fix $E \in [0, 1]$ and consider any $\epsilon \in [\frac{2}{3}, 1)$. Theorem 1 implies that*

$$\mathbb{P}^2(S_E) \geq E^2, \tag{9}$$

*where*

$$S_E \doteq \{(z_1, z_2) \in \mathbf{Z}^2 : \mathbb{P}(Z_3 \in \Gamma^\epsilon(z_1, z_2, z_1', ..., z_M')) \geq 1 - E\}.$$

*However, equation 9 does not provide a confidence interval for the probability of drawing a new sample in $Q$, or conversely in $\overline{Q}$. For $\epsilon \in [\frac{2}{3}, 1)$, $\Gamma^\epsilon = \mathbf{Z}$ with probability $b^2$ (from Case 3) and $\Gamma^\epsilon = \overline{Q}$ with probability $1 - b^2$ (from Case 1 and 2)[3], see Figure 1. Theorem 1 is a distribution-free result and as such it holds for all values of $b$, leading to two cases $b \leq E$ and $b > E$:*

1. *If $b \leq E$ (i.e. $1 - b \geq 1 - E$):*

    - *If $z_1, z_2 \in Q$ (Case 3) we have that $\mathbb{P}(Z_{N+1} \in \Gamma^\epsilon) = \mathbb{P}(\mathbf{Z}) = 1 \geq 1 - E$, hence $Q \times Q \subseteq S_E$.*
    - *If at least one of $z_1$ and $z_2$ belongs to $\overline{Q}$ (Case 1 and 2) we have that $\mathbb{P}(Z_{N+1} \in \Gamma^\epsilon) = \mathbb{P}(\overline{Q}) = 1 - b \geq 1 - E$, hence $\overline{Q \times Q} \subseteq S_E$.*
    - *Thus, $S_E = \mathbf{Z}^2$. Trivially, $\mathbb{P}^2(S_E) = \mathbb{P}^2(\mathbf{Z}^2) = 1 \geq E^2$.*

2. *If $b > E$ (i.e. $1 - b < 1 - E$)*

    - *If $z_1, z_2 \in Q$ (Case 3), as before, $\mathbb{P}(Z_{N+1} \in \Gamma^\epsilon) = \mathbb{P}(\mathbf{Z}) = 1 \geq 1 - E$, and once again $Q \times Q \subseteq S_E$.*
    - *If at least one of $z_1$ and $z_2$ belongs to $\overline{Q}$ (Case 1 and 2) then $\mathbb{P}(Z_{N+1} \in \Gamma^\epsilon) = \mathbb{P}(\overline{Q}) = 1 - b < 1 - E$, hence such $z_1$ and $z_2$ do not belong to $S_E$ by definition.*
    - *Thus, $\mathbb{P}^2(S_E) = \mathbb{P}^2(Q \times Q) = b^2 \geq E^2$.*

*Theorem 1 holds for both cases, since we have either $\mathbb{P}^2(\mathbf{Z}^2) = 1 \geq E^2$ or $\mathbb{P}^2(Q \times Q) = b^2 \geq E^2$. Now, assume $b > E$ and that the calibration set gives $R_1 = 0$ and $R_2 = 1$. What can we say about $b$?*

*For the given calibration set and significance level the INP predicts $\Gamma^\epsilon = \overline{Q}$, hence it is tempting to say that $\mathbb{P}^2(\mathbb{P}(\overline{Q}) \geq 1 - E) = \mathbb{P}^2(1 - b \geq 1 - E) \geq E^2$, or equivalently $\mathbb{P}^2(b \leq E) \geq E^2$: recalling equation 2, we may conclude that $[0, E]$ is a $E^2$ confidence interval for $b$. But this is clearly not true: since we assumed that $b > E$ the interval $[0, E]$ will never contain the parameter $b$ (note that none of the arguments of $\mathbb{P}^2()$ depends on $(z_1, z_2)$ in the preceding statement, unlike equation 2). We conclude from this example that this is not a viable path to obtain a PAC bound for $b$ comparable to equation 2.*

The example above leads us to the following remark and main message of this technical note.

**Remark 1** *Theorem 1 guarantees the correctness of the set predictor $\Gamma^\epsilon$. Adopting the frequentist perspective, it is a statement on how often the set predictor $\Gamma^\epsilon$ constructed from $N$ samples attains the desired coverage level $1 - E$ for a new realization of $Z_{N+1}$. In other words, since $b$ is unknown, if $b > E$ the INP attains the desired coverage level only when $\Gamma^\epsilon = \mathbf{Z}$ (which is a trivial prediction), and it does not attain the desired coverage level when $\Gamma^\epsilon = \overline{Q}$. Essentially, the confidence level of $E^2$ is attained by making trivial predictions sufficiently often. If instead $b \leq E$, the INP is always correct. Thus, Theorem 1 does not estimate $b$ or provide information on the probability of a specific score or class, which is the goal of BPCI methods. See the appendix for a graphical representation.*

To further clarify, consider the equivalent set predictor mapping the elements $z$ predicted by $\Gamma^\epsilon$ to their respective nonconformity score

$$\overline{\Gamma^\epsilon}(z_1, ..., z_N, z_1', ..., z_M') \doteq \bigcup_{z \in \Gamma^\epsilon(z_1, ..., z_N, z_1', ..., z_M')} A((z_1', ..., z_M'), z),$$

which amounts to $\overline{\Gamma^\epsilon} = \{0, 1\}$ when $\Gamma^\epsilon = \mathbf{Z}$ and $\overline{\Gamma^\epsilon} = \{0\}$ when $\Gamma^\epsilon = \overline{Q}$. Let $R_{N+1} \doteq A((z_1', ..., z_M'), Z_{N+1})$ be the score of the $N+1$-th sample. Then, we can replace the event $R_{N+1} \in \overline{\Gamma^\epsilon}$ with $Z_{N+1} \in \Gamma^\epsilon$ in equation 6. In essence, both BPCI methods and training-conditional CP provide PAC guarantees but differ in scope: while BPCI methods compute an interval containing the true value $b$ describing the probability of *the event that the $N + 1$-th score equals 1*, i.e. $R_{N+1} = 1$ (with probability not less than $1 - \alpha$), training-conditional CP computes a lower bound for the probability of *the event that the $N + 1$-th score is contained in the predicted set of scores*, i.e. $R_{N+1} \in \overline{\Gamma^\epsilon}$ (with probability not less than $1 - \delta$).

---

[3]If $\Gamma^\epsilon$ predicts $\overline{Q}$ it implies that the nonconformity score of $Z_{N+1}$ is predicted to be 0, whereas if it predicts $\mathbf{Z}$ then all we know is that the nonconformity score of $Z_{N+1}$ is predicted to be in $\{0, 1\}$ which is uninformative.

Remark 1 extends to any scenario where the nonconformity score takes values from a finite set, effectively defining a classification problem. Training-conditional CP provides a framework for constructing a set predictor that guarantees the desired coverage level with a minimum confidence. The predictor adapts to the calibration data: for 'good' calibration data, it produces tight sets (few classes), while for 'poor' calibration data, it outputs loose sets (many classes). On average, the probability that the calibration data yields a predictor attaining the coverage level of $1 - E$ is at least $1 - \delta$.

Depending on the choice of $\epsilon$ and $E$, we have shown that the $1-\delta$ confidence level may be achieved simply by predicting the entire sample space (i.e., all classes) sufficiently often (see Figure 2). However, this approach does not provide meaningful information about the probability of a specific class, which is the focus of equation 2 and, more generally, BPCI methods.

### 4.1  A Note on Safety Verification for Dynamical Systems

Recent studies have applied training-conditional CP, particularly Theorem 1, to provide PAC guarantees on the safety of control systems with neural network-based controllers (Chilakamarri et al., 2024; Lin & Bansal, 2024), and more broadly, on the safety of autonomous systems (Vincent et al., 2024). In this section we show that these works follow the reasoning outlined in section 4, and are therefore incorrect. Below, we follow the notation used in Lin & Bansal (2024), but the same applies to the other works.

Consider a dynamical system defined by $\dot{x} = f(x)$ where $x \in X \subseteq \mathbb{R}^n$, a fixed time horizon $T \in \mathbb{R}_{>0}$. Denote by $\xi_x(\tau)$ for $\tau \in [0, T]$ the state trajectory of the system at time $\tau$ when initialized at $x$ (for simplicity we assume that the solution to the differential equation exists and is unique)[4]. Let $X_A \subset X$ represent a set of undesirable states, and consider the cost function defined as

$$J(x) \doteq \min_{\tau \in [0,T]} d(\xi_x(\tau)),$$

where $d : X \to \mathbb{R}$ is a function satisfying

$$d(x) \leq \gamma \iff x \in X_A, \, d(x) > \gamma \iff x \in X \setminus X_A,$$

for some threshold $\gamma \in \mathbb{R}$. The function $d$ measures the distance between a point in the domain and the unsafe set $X_A$. An instructive example for the discussion is below is to choose $\gamma = 0$ and $d : X \to \{0, 1\}$, with $d = 0 \iff x \in X_A$ and $d = 1 \iff x \in X \setminus X_A$, but the same applies for any different choice. In this case, $J$ assigns a positive real number to a point $x \in X$ if and only if the state trajectory from $x$ never intersects with $X_A$. Let $(X, \mathcal{F}, \mathbb{P})$ be a probability space. To quantify system safety probabilistically, we seek to estimate $\mathbb{P}(\{x : J(x) > 0\})$, i.e. the probability of sampling an initial state that leads to a safe trajectory. In Lin & Bansal (2024) the authors define the nonconformity score as $R_i \doteq J(x_i)$ for $i = 1, ..., N$, and are therefore interested in estimating $\mathbb{P}(\{x : R^x > 0\})$. However, this is equivalent to defining a nonconformity measure as

$$A(x) \doteq \begin{cases} 1 \text{ if } x \in X_A, \\ 0 \text{ if } x \in X \setminus X_A, \end{cases} \tag{10}$$

and we have shown that this line of reasoning is not suitable for estimating the parameter $b$ of a Bernoulli r.v. given $N$ i.i.d. realizations $R_i \sim \text{Bern}_b$ of it.

Since Chilakamarri et al. (2024) relies on the framework of Lin & Bansal (2024), it suffers from the same issue. Additionally, in Vincent et al. (2024, Theorem 1), the authors re-derive Theorem 1, originally from Vovk (2012). They claim that training-conditional CP reduces to the Clopper-Pearson confidence interval when the underlying i.i.d. random variables are Bernoulli-distributed (see their Sec. Proofs-D). However, we have disproved this claim.

---

[4]In the original paper the trajectory $\xi$ depends on a learned controller and depends on a training set $Z'_1, ..., Z'_M$. For clarity we omit this dependence here, since the training set is given and is fixed.

## 5 Conclusion

In this note we examined existing methodologies to use training-conditional CP for statistical safety verification, a problem that can be reduced to estimating the expectation of a Bernoulli random variable. While training-conditional CP remains a powerful tool for uncertainty quantification we have shown that it is not appropriate for BPCI problems. Specifically, we clarified the correct interpretation of confidence intervals and PAC-style guarantees for training-conditional CP. We do not rule out the possibility that a different formulation of CP could be applied to BPCI problems. This is left for future work.

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

## A Appendix

We validate empirically equation 9 as follows and represent the results graphically in Figure 2.

We define a list of values for $E$ by $E_q = 0.01 + 0.01 * q$ for $q = 0, ..., 98$. For every value of $E_q$ we consider an underlying Bernoulli distribution with parameter $b_{1,q} = E_q - \alpha E_q < E_q$ (right figure) and an underlying Bernoulli distribution with parameter $b_{2,q} = E + \alpha E_q \% > E_q$ (left figure) with $\alpha = 0.005$. For every value of $q = 0, ..., 98$ we examine the two situations $b_{1,q} \leq E_q$ and $b_{2,q} > E_q$, as mentioned in Example 1 - Part 2. The significance level $\epsilon$ is set to $2/3$. We draw $n_{\text{cal}} = 5 \cdot 10^4$ pairs of calibration points $\{z_1^{(i)}, z_2^{(i)}\}_{i=1}^{n_{\text{cal}}}$. For every pair of calibration points $z_1^{(i)}, z_2^{(i)}$ we construct the resulting INP as $\Gamma_{(i)}^\epsilon \doteq \Gamma^\epsilon(z_1^{(i)}, z_2^{(i)}, ...)$, draw $n_{\text{test}} = 5 \cdot 10^4$ test points $\{z_{N+1}^{(j)}\}_{i=j}^{n_{\text{test}}}$ and compute the empirical frequency $\hat{g}_i = \frac{|\{j=1, ... n_{\text{test}}: z_{N+1}^{(j)} \in \Gamma_{(i)}^\epsilon\}|}{n_{\text{test}}}$ as an approximation for $\mathbb{P}(Z_{N+1} \in \Gamma_{(i)}^\epsilon)$; finally we compute $\hat{h} = \frac{|\{i=j, ... n_{\text{cal}}: \hat{g}_i \geq 1-E\}|}{n_{\text{cal}}}$ as an approximation to $\mathbb{P}^2(S_E)$ shown in the plots as the solid red line. The solid black line represents the curve given by $E^2$, which remains always below the red line in both plots, as expected. The area shaded in blue represents the fraction

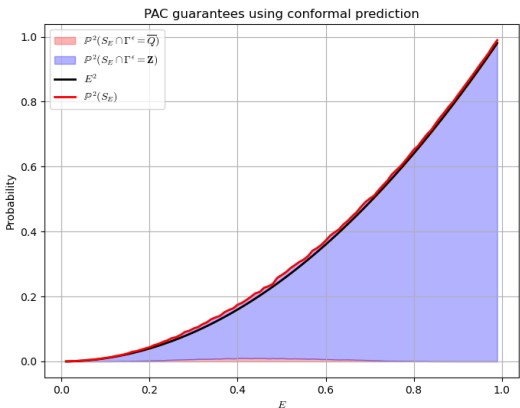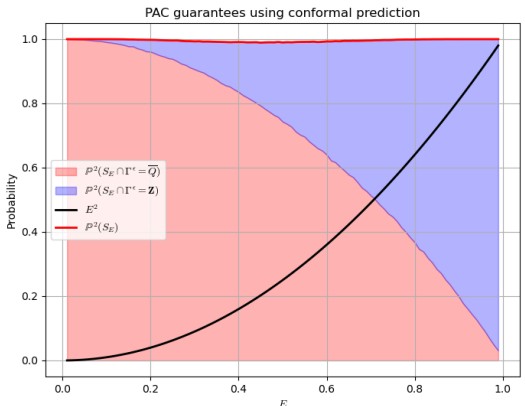

Figure 2: On the left the curves resulting from $b_{2,q} > E_q$, on the right the curves resulting from $b_{1,q} \leq E_q$, for $q = 0, ..., 98$.

of the $\hat{g}_i$'s for which the INP $\Gamma^\epsilon_{(i)}$ is equal to $\mathbf{Z}$, whereas the area shaded in red represents the fraction of the $\hat{g}_i$'s for which the INP $\Gamma^\epsilon_{(i)}$ is equal to $\overline{Q}$ *and $\hat{g}_i$ is greater or equal than $1 - E$.* It is visible in the left plot that the only reason why the solid red line (approximating $\mathbb{P}^2(S_{E_q}) = b^2_{2,q}$) is above $E^2_q$ is that the INP is allowed to predict the entire set $\mathbf{Z}$. In contrast, on the right the solid red line approximates $\mathbb{P}^2(S_{E_q}) = 1$ since any pair of $z_1^{(i)}, z_2^{(i)}$ results in a prediction $\Gamma^\epsilon_{(i)}$ satisfying $\mathbb{P}(Z_{N+1} \in \Gamma^\epsilon_{(i)}) \geq 1 - E_q$; accordingly, for a fixed $q$, the area shaded in red covers approximately $1 - b^2_{1,q}$ of the 'Probability' axis and the area shaded in blue approximately $b^2_{1,q}$.

In summary, in both situations the theorem is confirmed empirically, since the red line is always above the black line. In the first case, where $b_{2,q} > E_q$, the minimum confidence level of $E^2_q$ is attained by predicting sufficiently often the entire sample space $\mathbf{Z}$, precisely with a frequency of $b^2_{2,q}$, as this is the only set prediction attaining the required coverage probability of $1 - E_q$. Unfortunately, a prediction of the entire sample space is uninformative. In the second case, where $b_{1,q} \leq E_q$, any predicted set between $\overline{Q}$ and $\mathbf{Z}$ attains the required coverage probability of $1 - E_q$.

