# OpenReview forum: "On Training-Conditional Conformal Prediction and Binomial Proportion Confidence Intervals"
_TMLR — Accepted by TMLR_

### Review · Reviewer_N32P · 2024-12-12

**Summary Of Contributions:**

The authors show by counter-example that the usage of conformal prediction for providing safety certificates in learning-based control is not valid.

**Audience:**

Yes

**Broader Impact Concerns:**

No concerns.

**Claims And Evidence:**

Yes

**Requested Changes:**

1) Some abbreviations that the authors maybe consider general knowledge are not introduced, or at least not at the first use. For accessibility of the paper, I think it would be important to introduce those (e.g., i.i.d., r.v.). Similarly, at the beginning of Section 3, the filtration of the probability space appears, but the word "filtration" does not appear; this variable is not explained. I think also the use of superscripts for probabilities should be explained.

2) I think currently, in (4), the range of $i$ is not defined.

3) It would help to provide some more intuition on how we want to use the INP and what role it will play later in in the paper.

4) The example is nice and simple but not that simple to follow. Especially, the conclusion is not yet fully convincing. The bounds from Theorem 1 are probabilistic in nature. The confidence intervals hold with a certain probability. Thus, I may always construct a case where my confidence intervals are wrong. As long as this case has sufficiently low probability, this should not be a concern. I think in the end of the example, it should be made more precise what is the problem. Also in the discussion of the graphical example in the appendix, I don't think the point becomes sufficiently clear.

5) There seems to be some issue with the usage of \cite vs \citep. I would also recommend using the \secref commands provided in the TMLR papers - currently, it seems like the space between Sec. and the section number is too large.

**Strengths And Weaknesses:**

The paper's strength is that it makes an important point via a relatively simple example. However, the paper is extremely concise, making it hard to follow for people who are not experts on the topic but might think about leveraging conformal prediction. Exactly that audience would highly benefit from such a paper.

---

> ### Author Response · Authors · 2024-12-26
>
> We thank the reviewer for the constructive feedback and the time spent reviewing our submission. We hope to have addressed the concerns of the reviewer. To reduce the conciseness, we added some clarifications and paragraphs to improve the accessibility to non-experts. We have also added a reference to a tutorial on CP which includes applications of the theory to practical examples. In addition, we addressed the questions of the reviewer.
>
> 1.	We thank the reviewer for pointing this out. Accordingly, we have introduced the definitions of the abbreviations where needed.
> Regarding the comment on the filtration of the probability space, we assume that the reviewer refers to the $\mathcal{F}$ appearing in the definition of the probability space. We added a definition of each element, stating that $\mathbf{Z}$ is the sample space, calligraphic $\mathcal{F}$ is a $\sigma$-algebra on $\mathbf{Z}$, and $\mathbb{P}$ is a probability measure. We prefer using the terminology of $\sigma$-algebra for $\mathcal{F}$ and product $\sigma$-algebra for   $\mathcal{F}^N$ (appearing in Theorem 1) rather than the concept of filtration, following the usual terminology used in the conformal prediction literature.
> 2.	We have added the range of $i$ in equation (4).
> 3.	We have added a paragraph at the beginning of Section 3 to give some more intuition to the reader on the core ideas of conformal prediction and referenced some works containing applications of it. Additionally, we added an intuitive explanation of the role of the INP after its definition in equations (3)-(5). We hope that this provides the necessary intuition on the role the INP plays later in the paper.
> 4.	We thank the reviewer for pointing this out. We decided to split the example into two parts, added more explanations to guide the reader's intuition, supported by two new figures. We hope that this will make it easier to follow. We have also rephrased the main remark and numerical example in the appendix. The issue with using CP arises in Example 1 - Part 2: for instance if $b > E$ only a prediction of $\mathbf{Z}$ attains the necessary coverage probability (such prediction is uninformative for the value of $b$), and any time the INP predicts the set $\overline{Q}$, it produces the statement $\mathbb{P}(\overline{Q})=1-b\geq 1 - E$ which is false, similarly for the case of $b\leq E$. We are left with no information of the value of $b$.
> 5.	We have corrected the issue with \citep and \citet. However, we did not find the command \secref in the TMLR LaTeX stylefile. The stock main.tex file in TMLR template seems to be using the standard command \ref.
>
> We remain available for additional comments and requests for additional changes. Thank you.

---

> > ### Comment · Reviewer_N32P · 2025-01-01
> >
> > I thank the authors for the reply and revision of the paper. The .zip file that contains the TMLR LaTeX files also contains a file "math_commands.tex," which contains some definitions like \secref, etc. But this is clearly not a crucial point for the paper. In the revised version of the paper, there are some constructions like "don't" or "can't." I think the convention is not to use those in scientific writing. Apart from that, I think the additional explanations make the paper much clearer.

---

> > > ### Author Response · Authors · 2025-01-06
> > >
> > > We thank the reviewer for the feedback.
> > > We amended the manuscript according to the advice. We are now using \secref for referencing sections and removed constructs like 'don't' and 'can't'.
> > > We hope the reviewer is satisfied with the latest reviewed version of the manuscript.

---

### Review · Reviewer_xgpv · 2024-12-16

**Summary Of Contributions:**

In this paper, the authors address an issue with using the conformal predictions framework to solve the Binomial Proportion Confidence Intervals (BPCI) problem, specifically deriving bounds or confidence intervals for the estimated Bernoulli probability $b$. Through a simple toy example, the authors demonstrate that estimating a confidence interval is not feasible within the conformal prediction framework. They argue that this approach could lead to incorrect predictions because, although conformal predictions are based on Theorem 1 (Vovk, 2012), the results of this theorem cannot be directly applied to confidence interval estimation. Furthermore, the authors discuss the practical implications of their findings in the context of the Safety Verification for Dynamical Systems task. They highlight that recent works applying conformal predictions to this task are, in fact, incorrect.

**Audience:**

Yes

**Claims And Evidence:**

Yes

**Requested Changes:**

1. **[strengthen]** Example 1 seems slightly overloaded and hard to follow. I recommend dividing its different cases more distinctly. A complementary table showcasing the results (e.g., what INP equals in different cases) would be helpful.

2. **[strengthen]** Though the results of the toy example are appreciated and theoretically important, it is not entirely clear how this theoretical shortcoming of conformal prediction influences the real performance of systems. For instance, in [Vincent 2024], it was shown that the proposed intervals are useful—why is that the case here? A discussion would be much appreciated.

3. **[strengthen]** Building on the previous point, the provided example is a counterexample showing why conformal prediction might be inefficient in the considered scenario. Are there any assumptions under which conformal predictions could provide tight enough confidence intervals? A discussion on this would be valuable.

4. **[strengthen]** Could you elaborate on how the choice of INM influences the provided example? Is there a family of INMs that work for the BPCI problem, or is this a universal result?

**Strengths And Weaknesses:**

# Strengths

* The paper contributes to the field of conformal prediction and its applications to modern models and uncertainty estimation, a rapidly developing and important domain.

* The authors provide a clear example demonstrating their findings, which highlights the shortcomings of the conformal prediction framework within the context of the Binomial Proportion Confidence Intervals (BPCI) problem. Through this example, they effectively illustrate why confidence interval estimation is not feasible in this framework, underscoring its limitations.

# Weaknesses

(The following parts are discussed in more detail in the next section, "requested changes")

* Some parts of the presentation could be improved.
* The practical impact is unclear.
* Observing the described behavior in more challenging setups would be interesting.

---

> ### Author Response · Authors · 2024-12-26
>
> We thank the reviewer for the constructive feedback and the time spent reviewing our submission. We hope to have addressed the concerns of the reviewer. We improved the overall clarity of the document by adding new paragraphs with additional clarifications. Regarding more challenging setups, we generalize the implications of Example 1 in the paragraph before Section 4.1; there, we discuss a more general setting of classification problems. We focused on a simple example to help deliver the main message. Below, we address directly the questions raised by the reviewer.
>
> 1.	Following the reviewer’s advice, we split the example in two parts, we added a brief summary and a graphical representation of the INP to improve the clarity of the core message of the example and help the reader follow the derivation.
> 2.	Rather than highlighting a shortcoming of CP, in the note we highlight that CP has a different purpose than BPCI. CP is typically used to ‘upgrade’ a point predictor with no statistical guarantees, to a set predictor with statistical guarantees. This set can be a set of classes (for classification problems) or a set of real numbers (for regression problems) that with a guaranteed probability contain the correct prediction. In contrast, the purpose of BPCI is to provide an interval for the probability of the occurrence of one event, for example the occurrence of one class rather a different one: this is useful if one is interested in finding a confidence interval for the parameter $b$ of a Bernoulli distribution.
> In [Vincent 2024] the authors focus on applications of CP to safety. In the example where they use a binary nonconformity score they rely on Theorem 4 [Appendix D] therein which is the one-sided Clopper-Pearson method. They claim that if the nonconformity score can only take two values then CP returns a BPCI. This claim is only stated and not proved in [Appendix D]. Here we are disproving their claim. Moreover, we are stressing that CP has a different meaning than BPCI. Depending on the problem at hand one may be interested in applying CP or BPCI. These two methods are not directly comparable to say that one is better than the other one.
> 3.	The counterexample highlights that CP can not be used to derive a BPCI for a Bernoulli random variable, something claimed by the works cited in Section 4.1. To be precise, this is not a matter of inefficiency of CP: it is rather that CP has a different purpose. While equation (2) appears to be similar to equations (6) and (7), their meaning is very different, as we mention in Remark 1. It is worth mentioning that CP provides efficient (tight) prediction sets when the nonconformal scores have a continuous distribution, such as in regression problems [Angelopoulos 2023].
> 4.	If we restrict the INM to any ‘indicator-like’ function mapping points in the sample space Z to any two real values (or more generally to a finite number of values) then the result is universal. This is the case for classification problems. BPCI methods provide a way to compute a confidence interval for the probability of occurrence of one specific class. Instead, CP was developed with the idea of accepting set predictions with statistical guarantees, rather than single-class predictions with no statistical guarantees. To the best of our knowledge, we are not aware of a formulation of CP for statistical guarantees on single-class predictions, in a similar fashion to BPCI.
>
> We remain available for additional comments and requests for additional changes. Thank you.

---

> > ### Comment · Reviewer_xgpv · 2025-01-20
> >
> > I appreciate the authors' response and revisions, which have significantly improved the paper's clarity. The additional explanations effectively address my questions and concerns. The paper now reads well to me.

---

### Review · Reviewer_R9Jz · 2024-12-17

**Summary Of Contributions:**

The paper identifies a significant flaw that has been repeated in the literature. Specifically, several works have used training-conditional conformal prediction (CP), particularly in the form of Theorem 1, to provide PAC guarantees on the safety of dynamical systems. However, the theorem has been misinterpreted and applied incorrectly in scenarios where it is unsuitable, leading to problematic implications.

**Audience:**

Yes

**Claims And Evidence:**

Yes

**Requested Changes:**

See the weakness above and please answer the question.

**Strengths And Weaknesses:**

# Strengths

The work is highly relevant to the TMLR community. It highlights an important issue in the misuse of training-conditional conformal prediction with concrete examples, serving as a crucial warning that could have significant impact on future research.

# Weaknesses

The paper generally provides the necessary material to support its points, but it could benefit from clearer communication through more intuitive explanations and higher-level walk-throughs.

- For instance, Example 1 is quite dense. It might be clearer by having a brief top-down explanation of how the example will demonstrate the point could help readers better anticipate what to expect. Also, you can perhaps start with something like, “We give an XXX answer with the following example,” rather than “We proceed with an example.”

- The paper could also improve clarity by adding interpretations after definitions, explaining what high or low values of the terms (e.g., R^z, p^z) imply. For example, in Eq. (8), is it correct to interpret Q as the set of samples that do not conform? If so, clarifying this interpretation when introducing Q would help. Similarly, providing a verbal, intuitive explanation of the function d would make the discussion more accessible.

- There are quite some similar notations about Binomial distributions and Bernoulli distributions which might bring some confusion. Also, $Bin_{N,E}(\cdot)$ in Theorem 1 needs a definition.

- The parameter $b$ mentioned just before Example 1, seems to appear abruptly. It would help to specify that b is the probability of $z\in Q$ when the training set is fixed.

# Questions

1. In Theorem 1, do you need to fix $N$ before discussing the event? Also, this result also depends on the function $A$., so it might be clearer to explicitly include A in the statement.

2. Could you elaborate more on what you mean by Theorem 1 being a distribution-free result? Does it simply mean that the result is independent of b? If so, it might be helpful to state this more explicitly for clarity.

3. Regarding Eq. (9), I do not find the conclusion immediately straightforward. If Theorem 1 were applied directly, shouldn’t the result actually be a Binomial tail probability? The conclusion about $E^2$ seems to arise only from the argument that follows Eq. (9).

4. I may have missed something, but there is still an aspect of Example 1 that is unclear to me. Specifically, when the training set is fixed, $Q$ and $b$, as defined through $A$, are mutually dependent and both dependent on the training set. Meanwhile, $p^z$ and $\Gamma^\epsilon$ also depend on the calibration set. How, then, can you discuss scenarios with different values of $b$ while seeming to hold the set $\bar{Q}$ fixed? Could you clarify this part?

---

> ### Author Response · Authors · 2024-12-26
>
> We thank the reviewer for the constructive feedback and the time spent reviewing our submission. We hope to have addressed the concerns of the reviewer.
>
> We introduced an additional high-level explanation at the beginning of Section 3 on the idea of CP and before Theorem 1 on the role of the INP.
>
> We split the example into two parts and added a new figure to guide the reader's intuition. We also anticipate the conclusion before Example 1 and at the beginning of Example 1 - Part 2.
>
> We added interpretations of the main definitions, for instance on the role of the INP after equations (3)-(5). We also added an interpretation of $Q$ as the set of points that do not conform and an interpretation of $d$ as a distance function.
>
> We thank the reviewer for noticing this. We have adjusted the notation for the binomial distribution to make it consistent in all the sections. We have also added a definition of $\text{Bin}_{N,E}()$ in Theorem 1.
>
> We added a definition before Example 1 of the parameter $b$.
>
> Below we address the remaining questions.
>
> 1.	Following the reviewer’s advice, we added the requested details to the statement of Theorem 1.
> 2.	For a result to be distribution-free it is meant in general that it holds for any probability distribution, as long as the samples used are i.i.d. and distributed according to the underlying distribution of interest. In our case, this means indeed that it holds for any Bernoulli-distributed set of samples, for any value of the parameter $b$. Following the reviewer’s advice, this is now stated more clearly after Theorem 1. Additionally, we stress this point again in Example 1 – Part 2, where we make use of it.
> 3.	If we understand correctly the question, the reviewer is asking whether $E^2$ is the tail of a binomial distribution. The answer is yes. For $N = 2$, $\epsilon \in [2/3, 1)$, from the definition of $\delta$ in Theorem 1, we have $\delta = \text{Bin}_{2,E}(1)$, which equals $1-E^2$. Then the left-hand side of equation (7) gives $1-\delta=E^2$, which is the tail of a binomial distribution with 2 trials and probability of success $E$. We remain available for further clarification.
> 4.	Typically, the set $Q$ is dependent on the training set. Here we are discussing the case where the nonconformity score can only take two values, in particular, it maps points in $Q$ and $\overline{Q}$ to two real values. The set $Q$ in turn defines the value of the parameter $b$. However, this value is unknown, therefore we discuss which situations arise for different values of $b$. The goal is to gather information on the value of $b$ in the CP framework using a calibration set. In particular, we show that in the CP framework it is not possible to recover any useful estimate on the value of $b$.
>
> We remain available for additional comments and requests for additional changes. Thank you.

---

> > ### Comment · Reviewer_R9Jz · 2025-01-07
> >
> > Thanks for the clarification and the revision. It does read clearer now. However, I believe Reviewer N32P raises a valid point. My personal guess is that people who make mistakes in interpreting these results might be confused by the various probability statements and the two PAC guarantees. Overall, in addition to saying X and Y are different and explaining their individual meanings, it could be helpful to make more direct comparisons on how exactly they should be different.
> >
> > For example, while you do mention that the names of $\epsilon$, $E$, and $\alpha$ are different, it could be helpful to point out the difference more explicitly. Also, at the beginning of Section 3, you use $\epsilon$ as the "desired error probability," while later it represents the "significance level." Additionally, when discussing in Section 3 the nested statement as a PAC statement, it might help to explicitly address potential confusion regarding how this PAC statement differs from another PAC guarantee.
> >
> > While the concrete example in Section 4.1 is valuable, ensuring that readers already understand what went wrong before encountering the example would strengthen the paper’s clarity and impact. I hope this makes sense.

---

> > > ### Author Response · Authors · 2025-01-08
> > >
> > > We thank the reviewer for their feedback.
> > >
> > > It is not entirely clear what the reviewer means by 'how exactly they $\textbf{should}$ be different'. We hope we have understood correctly the comment.
> > >
> > > The apparent discrepancy between the terms "desired error probability" and "significance level" is because in the original formulation of CP $\epsilon$ quantifies both the error probability and the significance level. In the training-conditional formulation, this is not the case. The beginning of Section 3 was modified after the request of N32P to provide some high-level explanation of CP, which is best understood as referring to the ideas of the original formulation of CP.
> > > To address the reviewer's feedback we have added details regarding $\epsilon$ and $E$ (footnote on page 3) and a comparison between $\delta$ and $\alpha$ (final part of the paragraph before Section 4). We have followed the original notation and terminology introduced by Vovk. We hope that with these additional details, the meaning of the notation is clearer.
> > >
> > > We have dedicated section 3 to give a self-contained introduction to training-conditional CP, where we provide an interpretation of the PAC guarantees. The reason why in section 3 we do not discuss the difference with the PAC guarantees of section 2 is that we need to consider first the special case where the nonconformity score is given by an indicator function. We do so in Section 4 and we immediately jump to comparing sections 2 and 3. Since the difference between the two PAC guarantees is subtle, as can be seen by the publications that have confused the two, explaining such a difference without a concrete example at hand might not be as effective. There isn't something that went wrong: the two approaches simply have different meanings, and we highlight this in the situation considered in Section 4. To address the reviewer's feedback we added more details before Example 1 - Part 1 where we state that the setting considered is solvable by BPCI methods and it is not solvable using training-conditional CP. This should help the reader to better anticipate the conclusion of the example. Additionally, to better highlight the difference between the PAC statements we added a paragraph under Remark 1, where we discuss the general setting with $N$ calibration points. This should corroborate the reader's understanding of section 4 and the differences between the PAC guarantees.
> > >
> > > In case we have misunderstood the comment we apologise and we remain available for additional feedback and clarifications.

---

> > > > ### Comment · Reviewer_R9Jz · 2025-01-09
> > > >
> > > > Thanks for the prompt revision. The paper reads clear to me now.

---

### Comment · Action_Editor_qvBy · 2024-12-30
**Please have a look at the author responses**

Dear reviewers,

Thanks a lot for reviewing this paper!

The authors have submitted their official responses. Please have a look at them. If you have any follow-up questions, please ask by replying to their responses.

Thanks!

---

### Decision · Action_Editor_qvBy · 2025-02-05

**Recommendation:** Accept with minor revision

**Comment:**

All the three reviewers recommend to accept this paper. After reading the paper, the reviews, and the author responses, I agree with their recommendations.

My only minor suggestion is that the authors should polish the writing of Section 4 a little bit to make it more readable.

**Audience:**

Yes

**Claims And Evidence:**

Yes, all the reviewers and I think that the claims made in the submission are supported by accurate, convincing, and clear evidence.

---

> ### Author Response · Authors · 2025-02-10
> **Authors' Response**
>
> We thank the AE and the reviewers for their valuable and constructive feedback during the revision process of this paper.
> In response to the AE's comment, we have improved the readability of Section 4 by simplifying or rephrasing some sentences, subdividing certain paragraphs and enumerating the takeaway points in the example while preserving the additional content suggested previously by the reviewers. We hope that the final version satisfies the AE and the reviewers.